# Biomarkers Associated with the Outcome of Traumatic Brain Injury Patients

**DOI:** 10.3390/brainsci7110142

**Published:** 2017-10-27

**Authors:** Leonardo Lorente

**Affiliations:** Intensive Care Unit, Hospital Universitario de Canarias, La Laguna, Santa Cruz de Tenerife 38320, Spain; lorentemartin@msn.com; Tel.: +34-922-678-000

**Keywords:** biomarkers, prognosis, mortality, traumatic brain injury, caspase-3, total antioxidant capacity, melatonin, malondialdehyde, substance P

## Abstract

This review focuses on biomarkers associated with the outcome of traumatic brain injury (TBI) patients, such as caspase-3; total antioxidant capacity; melatonin; S100B protein; glial fibrillary acidic protein (GFAP); glutamate; lactate; brain-derived neurotrophic factor (BDNF); substance P; neuron-specific enolase (NSE); ubiquitin carboxy-terminal hydrolase L-1 (UCH-L1); tau; decanoic acid; and octanoic acid.

## 1. Introduction

Traumatic brain injury (TBI) results in many deaths, disabilities, and the consumption of healthcare resources each year [1]. All these facts have motivated interest in research on prognostic biomarkers in TBI patients [2,3,4,5,6,7,8,9,10]. This review focuses on biomarkers associated with the outcome of TBI patients, such as caspase-3, total antioxidant capacity, melatonin, S100B protein, glial fibrillary acidic protein (GFAP), glutamate, lactate, brain-derived neurotrophic factor (BDNF), substance P, neuron-specific enolase (NSE), ubiquitin carboxy-terminal hydrolase L-1 (UCH-L1), tau, decanoic acid, and octanoic acid. The interest in these biomarkers lies in that they could be used as prognostic biomarkers and that the modulation of some of them could open new research lines in the treatment of TBI patients to reduce the risk of mortality.

## 2. Biomarkers

### 2.1. Caspase-3

Cell death by apoptosis occurs mainly through the activation of the extrinsic pathway (death receptor pathway in type I cells) or intrinsic pathway (mitochondrial pathway in type II cells) [11,12,13,14,15]. Apoptosis begins in type I cells when the union between the surface death receptor of tumor necrosis factor receptor superfamily (TNFRSF) and the ligand of tumor necrosis factor superfamily (TNFSF) occurs, which produces the appearance of a death signal and causes pro-caspase-8 to be cleaved in active caspase-8, resulting in the activation of caspase-3. Apoptosis begins in type II cells by the action of pro-inflammatory cytokines such as interleukin (IL)-1 and IL-6, and of oxygen free radicals; afterwards, cytochrome-c is released from mitochondria to cytosol and caspase-3 is activated. Therefore, in both pathways of apoptosis (intrinsic and extrinsic pathways), caspase-3 is activated, and death cell occurs. 

In brain tissue samples from animals [16,17,18] and humans [19,20], apoptotic changes have been found after a TBI; also, in brain tissues from animal models of TBI [21,22,23,24,25,26,27,28,29], an increase of caspase-3 has been found [21,22,23,24,25,26,27,28,29]. Higher caspase-3 levels in TBI patients compared to controls in cerebrospinal fluid [30,31,32] and in brain tissue [19,33] have been found. In addition, higher caspase-3 levels in the brain tissue (periischemic zone of traumatic cerebral contusions) of non-surviving compared to surviving TBI patients have been found [33]. In a study by our team, we found that there is an association between serum levels of caspase-3 and TBI mortality [34]. I think that these higher serum caspase-3 levels in non-survivor patients could lead to a higher degree of apoptosis.

### 2.2. Total Antioxidant Capacity (TAC)

An increase in reactive oxygen species (ROS) production occurs in TBI and is involved in the development of secondary brain injury [35,36,37,38], by means of cellular dysfunction, microvascular regulation loss, vasogenic edema formation, and post-traumatic ischemia. Antioxidants attempt to compensate this increased ROS production to avoid oxidation. TAC determination may give relevant information about the antioxidant status [39,40].

In two studies, significant differences were not found in circulating TAC levels between non-survivor and survivor patients [41,42], and in three studies higher circulating TAC levels were found in non-survivor compared to survivor patients [43,44,45]. In a study published by our team, an association between serum TAC and mortality in TBI patients was found, as well as an association between serum levels of TAC and TBI severity assessed by the Acute Physiology and Chronic Health Evaluation-II score and the Glasgow Coma Scale, and a positive association between serum levels of TAC and malondialdehyde (MDA) levels [44]. In another study, higher serum TAC levels at admission, 24 h, and 48 h were found in patients with a poor functional outcome at six months post-TBI, in comparison to patients with a good functional outcome [45].

The increase of ROS leads to lipid peroxidation, and MDA is an end-product of this lipid peroxidation due to cellular membrane phospholipids degradation [46,47]. MDA is released into extracellular space and afterwards appears into the blood. The determination of circulating MDA levels has been used in other clinical circumstances, such as sepsis [48,49], brain infarction [50], or hepatocellular carcinoma [51], as an effective biomarker of lipid oxidation. Higher levels of MDA have been found in patients with TBI compared to healthy controls [9,52,53,54,55,56]. In addition, higher levels of MDA were found in non-surviving compared to surviving TBI patients in erythrocytes [55,56] or serum [9,57]. Moreover, in a previous study by our team [9], an association was found between serum MDA levels and mortality in TBI patients.

I believe that these increased serum TAC levels in non-survivor patients compared to survivors may be due to an attempt to compensate the higher free radicals production and higher peroxidation (according to the higher serum MDA concentrations); however, this increase in TAC is not enough to compensate for this unfavorable clinical state.

### 2.3. Melatonin

Melatonin is synthesized with a circadian rhythm (with higher production during the night than during the day) in the pineal gland, and also without circadian rhythm in other organs such as bone marrow, retina, gastrointestinal tract, and thymus [58]. Melatonin, in addition to participating in sleep regulation [59], could have antioxidant, anti-inflammatory, and antiapoptotic effects [60,61,62,63,64,65,66,67,68,69].

In some studies, lower concentrations of melatonin were found in TBI patients compared to healthy controls in saliva [70] or serum [71,72,73]; however, in one study, higher levels of melatonin were found in TBI patients compared to healthy controls in cerebrospinal fluid [74]. In a study recently published by our team, an association between serum levels of melatonin and mortality in TBI patients, as well as a positive association of serum levels of melatonin with serum TAC levels and with serum MDA levels were found [75]. 

I think that those increased serum melatonin levels in non-survivor TBI patients compared to survivor patients could be due to an attempt to compensate for the unfavorable clinical situation with higher ROS production and higher oxidant states (assessed by increased serum levels of MDA); however, this compensation is not achieved, and finally the death of the patient occurs.

### 2.4. S100B Protein

This protein has been found in astrocytes and Schwann cells, and also has been detected in other tissues such as melanocytes, bone marrow cells, lymphocytes, chondrocytes, and adipocytes. S100B protein plays a role in the regulation of intracellular levels of calcium and is eliminated by kidneys [76,77]. In some studies, higher circulating S100B levels were found in non-survivor compared to survivor TBI patients [78,79,80,81,82,83,84]. It could be possible that those high levels in non-survivor patients indicate the expression of astrocyte damage.

### 2.5. Glial Fibrillary Acidic Protein (GFAP)

This is the main protein of cytoeskeletal filaments in astrocytes, although it is also present in Schwann cells, spleen, and bone marrow. Higher circulating GFAP levels in non-survivor than in survivor TBI patients have been found [85,86,87]. Those GFAP levels in non-survivor patients could represent higher astrocyte damage. 

### 2.6. Glutamate

Glutamate is the main excitatory amino acid in humans, and high glutamine levels in cerebral microdialysis have been associated with a poor prognosis in TBI patients [88,89,90]. Recently, one study found higher glutamate levels in cerebrospinal fluid in non-survivor compared to survivor TBI patients [91]. Glutamate at high concentrations could be toxic to neurons involving an excessive influx of Na^+^ and Ca2^+^, mitochondrial dysfunction, and apoptosis [90].

### 2.7. Lactate

During brain hypoxia, an increase of lactate levels and lactate/pyruvate ratio appears to maintain energy production [92,93]; thus, high lactate levels could represent hypoxia in TBI patients. High levels of lactate have been found in TBI patients with unfavorable outcome in microdialysis [94], as well as in cerebrospinal fluid [91,95,96]. 

### 2.8. Brain-Derived Neurotrophic Factor (BDNF)

BDNF is a neurotrophin that protect neurons against glutamate excitotoxicity [97]. Studies on BDNF levels in TBI patients have been scarce, and their results have been controversial; thus, the role in TBI remains unclear. On the one hand, an association between TBI patient outcome and BDNF levels in serum [98] or cerebrospinal fluid [91] have not been found. On the other hand, an association between poor TBI patient outcome and low BDNF levels in serum [99] as well as high BDNF levels in cerebrospinal fluid [100] have been found.

### 2.9. Substance P

Substance P is a neuropeptide included in the tachykinin family that is mainly synthesized in the central and peripheral nervous systems [101]. Substance P exhibits pro-inflammatory effects, and has been associated with vascular permeability, development of brain edema, and functional deficits after TBI in an animal model [102].

Increased Substance P (SP) in brain tissue samples from TBI patients with neuropathological abnormalities compared with patients without abnormalities have been found [103]. In a study by our team, we found that serum SP levels were associated with TBI severity as well as with early mortality [8]. 

### 2.10. Neuron-Specific Enolase (NSE)

NSE is an enzyme present in the cytoplasm of neurons that participates in the glycolysis pathway. An association between high circulating levels of NSE and unfavorable outcome in TBI patients has been found [104,105,106,107,108]. Those high NSE levels in unfavorable outcome patients could represent greater damage of neurons.

### 2.11. Ubiquitin Carboxy-Terminal Hydrolase L-1 (UCH-L1)

UCH-L1 is an enzyme present in the soma of neurons. Higher circulating levels of UCH-L1 has been found in non-survivor compared to survivor TBI patients [109,110], and may be the expression of neuron damage.

### 2.12. Tau

Tau is a protein of the microtubule-associated protein (MAP) family that stabilizes microtubular assembly in the axons of neurons. High Tau levels in cerebrospinal fluid has been associated with a poor outcome in TBI patients [111]. Those tau levels in non-survivor patients could represent higher astrocyte damage. 

### 2.13. Metabolomics

An alteration in brain metabolism appears in TBI, and metabolomics (that consist of a large-scale study of metabolites in a biosample) could help to better our knowledge of TBI pathophysiology [112]. In a recently published metabolomics study, 465 metabolites were determined in serum samples from 211 TBI patients [113]. Out of 465 metabolites, 49 were significantly different between patients with unfavorable and favorable outcomes, most of them upregulated in poor outcome patients. For example, high levels of decanoic acid and octanoic acid (both medium-chain fatty acids) were associated with unfavorable outcome; also, both medium-chain fatty acids have been associated with mitochondrial dysfunction [114].

## 3. Future

The interest in these biomarkers lies in that they could be used as prognostic biomarkers. I think that the evidence that we have at present about these biomarkers will not change the daily clinical practice, and that the sole use of these biomarkers should be taken with caution; however, their use could help in the mortality prediction of TBI patients estimated by other prognostic factors (such as APACHE-II, Glasgow Coma Scale (GCS), age, and computer tomography findings).

Another point of interest concerning these biomarkers is the fact that is possible that the modulation of some of these biomarkers by the administration of different agents could incite research on new lines in the treatment of patients with TBI. In respect to caspase-3, the administration of caspase-3 inhibitors has reduced caspase-3 activity and apoptosis in brain tissues in rat models [24,25,26,27,28,29]. Regarding the total antioxidant capacity, a reduction of MDA have been found with the administration of different antioxidant agents (memantine, amantadine sulphate, melatonin); for example, a reduction of MDA levels in brain tissues has been achieved in animal models with the administration of memantine [115], as well as a reduction of circulating MDA levels with the administration of amantadine sulphate in TBI patients [116], and a reduction of MDA levels in brain tissues in animal models with the administration of melatonin [117,118]. With respect to melatonin, the administration of melatonin in animal models has led to antioxidant effects [117,118,119,120,121,122,123], anti-apoptotic effects [123], a reduction in brain edema [121,122,124], and anti-inflammatory effects [120,125]. With respect to substance P, the administration of an antagonist of neurokinin-1 receptor (which is the receptor of substance P) has reduced substance P activity and brain edema formation, and improved functional outcome [102].

## 4. Conclusions

Circulating levels of caspase-3, total antioxidant capacity, melatonin, S100B protein, glial fibrillary acidic protein, brain-derived neurotrophic factor (BDNF), substance P, neuron-specific enolase (NSE), ubiquitin carboxy-terminal hydrolase L-1 (UCH-L1), decanoic acid, and octanoic acid, as well as levels in microdialysis and cerebrospinal fluid of glutamate, lactate, and cerebrospinal fluid tau levels have been found to be associated with the outcome of TBI patients. These biomarkers could help in the prognostic classification of TBI patients and could open new research lines in the treatment of patients with TBI.

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
