# Peer review of "Biomarkers Associated with the Outcome of Traumatic Brain Injury Patients"

_brainsci, 2017, doi:10.3390/brainsci7110142_

Round 1

Reviewer 1 Report

The paper needs to be re-written to be more clear.

Regarding the grammar errors in the manuscript.  The paper would benefit from having it thoroughly edited and re-submitted.  

For instance in the abstract: "Antioxidants establish complex interactions"   that statement is meaningless, yet it is used as the rationale to measure TAC.  The Melatonin addition should be incorporated in a more seamless manner.  The measurement seems like an afterthought.  "TBI produces a large quantity..."  This is a poor word choice, especially for the opening sentence TBI results in...

The introduction paragraph repeats verbatim the same mistakes that were in the abstract.

Serum Caspase

Several run-on sentences in this paragraph. lines 33-35 and 35-38

line 31 omit pathway or by

line 32 omit activation of

line 44 omit besides have been found

line 45 "in a study were found higher ..."  What does this mean?

These were the issues I found in the first couple paragraphs.   I did not continue to read the article at this point because I feel the scientific meaning was being lost in the writing mistakes.  I felt as a consequence of the writing my data interpretation and analysis may be incorrect.

I hope this sheds light on my issue with the paper.

Author Response

The paper have been re-written.

In addition, I have included other biomarkers.

Reviewer 2 Report

The author reviews recent data related biomarkers to outcome in traumatic brain injury. The review focuses primarily on the author's own research, which includes Capase-3, total anti-oxidant burden, and melatonin. The author does not include material related other markers, such as glutamate, lactate, plasma metabolomies, S100B, GFAP, BDNF, and substance P, among others. Inclusion of summary of those biomarkers judged to be most important and addressing the shortcomings of others would make the work much more meaningful.

Some English language and other editing would also be helpful. For instance, on page 2, line 48-51 has some redundancy in the text.

Also, although mostly a stylistic comment, the author (a single author) uses first person plural for the pronoun as the subject or modifier of his work.

Author Response

The author reviews recent data related biomarkers to outcome in traumatic brain injury. The review focuses primarily on the author's own research, which includes Capase-3, total anti-oxidant burden, and melatonin. The author does not include material related other markers, such as glutamate, lactate, plasma metabolomies, S100B, GFAP, BDNF, and substance P, among others. Inclusion of summary of those biomarkers judged to be most important and addressing the shortcomings of others would make the work much more meaningful.

In the new version of the manuscript all those biomarkers have been included

Some English language and other editing would also be helpful. For instance, on page 2, line 48-51 has some redundancy in the text.

The paper have been re-written.

Also, although mostly a stylistic comment, the author (a single author) uses first person plural for the pronoun as the subject or modifier of his work.

I have corrected that.

Round 2

Reviewer 1 Report

This paper includes more biomarkers, but some are more developed than others.  The recently added biomarkers appear as afterthoughts. The DA and OA section reads more like metabolomics than the actual compounds. Each biomarkers should include a hypothesis of how it might fit into the mechanism of affecting TBI.  The introduction repeats exactly the abstract, which does not make the abstract a summary of the information.   This paper overall should be developed.

The writing still needs a lot of work.  For example, preposition choices, "have and has found" , redundancy, misspellings, subject-noun agreement.

line 117 "...protein has been found ovell all in astrocytes"    This type of lack of editing mistake is egregious.   

Author Response

This paper includes more biomarkers, but some are more developed than others. The recently added biomarkers appear as afterthoughts.

As was suggested by the reviewer, some are more developed than others. We have developed more those biomarkers that could be modulated by the administration of agents,

The DA and OA section reads more like metabolomics than the actual compounds.

As was suggested by the reviewer, we have changed that section as metabolomic.

Each biomarkers should include a hypothesis of how it might fit into the mechanism of affecting TBI.

As was suggested by the reviewer, we have included a hypothesis for each biomarker.

The introduction repeats exactly the abstract, which does not make the abstract a summary of the information.

As was suggested by the reviewer, we have modified the introduction section.

This paper overall should be developed.

As was suggested by the reviewer, the paper has been overall developed.

The writing still needs a lot of work.  For example, preposition choices, "have and has found" , redundancy, misspellings, subject-noun agreement.

line 117 "...protein has been found ovell all in astrocytes" This type of lack of editing mistake is egregious.

We have modified the English language

Reviewer 2 Report

The authors have addressed the reviewers comments satisfactorily. 

Author Response

Thanks for your positive comment.